# A Predictive Quality Inspection Framework for the Manufacturing Process in the Context of Industry 4.0

**DOI:** 10.3390/s24175644

**Published:** 2024-08-30

**Authors:** Stefan Rydzi, Barbora Zahradnikova, Zuzana Sutova, Matus Ravas, Dominik Hornacek, Pavol Tanuska

**Affiliations:** 1Faculty of Materials Science and Technology in Trnava, Institute of Applied Informatics, Automation and Mechatronics, Slovak University of Technology in Bratislava, 811 07 Bratislava, Slovakia; dominik.hornacek@stuba.sk (D.H.); pavol.tanuska@stuba.sk (P.T.); 2PredictiveDataScience, s.r.o., Klzava 31, 831 01 Bratislava, Slovakia; barborka.zahradnikova@predictivedatascience.sk (B.Z.); zuzana.sutova@gmail.com (Z.S.); matus.ravas@predictivedatascience.sk (M.R.)

**Keywords:** automotive manufacturing, quality control, defect detection, AI framework, production efficiency, predictive maintenance, data analytics, predictive quality inspection, machine learning

## Abstract

The purpose of this research is to develop an innovative software framework with AI capabilities to predict the quality of automobiles at the end of the production line. By utilizing machine learning techniques, this framework aims to prevent defective vehicles from reaching customers, thus enhancing production efficiency, reducing costs, and shortening the manufacturing time of automobiles. The principal results demonstrate that the predictive quality inspection framework significantly improves defect detection and supports personalized road tests. The major conclusions indicate that integrating AI into quality control processes offers a sustainable, long-term solution for continuous improvement in automotive manufacturing, ultimately increasing overall production efficiency. The economic benefit of our solution is significant. Currently, a final test drive takes 10–30 min, depending on the car model. If 200,000–300,000 cars are produced annually and our data prediction of quality saves 10 percent of test drives with test drivers, this represents a minimum annual saving of 200,000 production minutes.

## 1. Introduction

In recent years, advancements in technology have significantly impacted the manufacturing industry, particularly in enhancing quality control (QC) and quality management (QM). These advancements have led to the development of sophisticated predictive maintenance solutions aimed at improving the efficiency and reliability of industrial equipment. Traditional QC and QM methods, such as Six Sigma and Lean Manufacturing, focus on reducing variability and improving process efficiency [1,2]. However, there is a growing need for more advanced methodologies that leverage real-time data analysis and AI capabilities to predict quality issues in manufacturing processes [2,3].

### 1.1. Existing Research Methods

Traditional methodologies in QC and QM often involve manual inspections and standard statistical techniques. The Cross-Industry Standard Process for Data Mining (CRISP-DM) [4] is a widely adopted framework for implementing predictive strategies in quality inspection. These methodologies include steps like business understanding, data understanding, data preparation, model training, evaluation, deployment, and regular reassessment.

Despite the effectiveness of traditional methodologies, they often fall short of addressing the dynamic nature of modern manufacturing processes. Traditional methods may not adequately support continuous improvement and real-time data analysis, which are crucial for maintaining high-quality production standards. Additionally, the lack of integration with advanced AI techniques limits the ability to provide actionable insights promptly and accurately [5].

### 1.2. Problems to Be Solved by the Proposed Methodology

The proposed project seeks to address the limitations of existing QC and QM methods by implementing a more agile and iterative approach to predictive quality inspection [6]. By integrating AI into these processes, the framework will enable real-time analysis and decision-making to provide accurate, quality predictions [7]. This integration ensures continuous improvements in production processes within automotive manufacturing plants, offering a sustainable, long-term solution for predicting automobile quality at the end of the production line. The principal results show significant improvements in defect detection and support personalized road tests, thus enhancing production efficiency and reducing costs.

Another important goal of the implementation was to address the constant change in defect types as production quality managers eliminated detected defects in subsequent productions. This meant that while old defects were no longer occurring, new defect types were appearing. We solved the problem by designing and implementing an automated self-learning machine learning framework (AutoML) that runs at defined time intervals.

The AutoML framework implemented in our current project includes capabilities for model selection, hyperparameter optimization, and continuous learning. By automating these processes, the framework ensures that the most appropriate machine learning models are always used, even as production conditions and defect characteristics change. This approach not only improves the accuracy of defect detection but also significantly reduces the time required to update and deploy new models, improving the overall efficiency of the production line.

In addition, AutoML facilitates the integration of diverse data sources, enabling the system to utilize a wide range of input variables from different stages of the manufacturing process. In one of our previous automotive projects, the AutoML system was instrumental in correlating data from multiple sensors and production stages, resulting in the early detection of subtle defect patterns that would have been difficult to identify using traditional methods. By using AutoML, we have been able to create a robust, adaptive system that continuously improves its predictive accuracy and relevance, ensuring that the quality inspection process is more effective.

## 2. The Background of the Study

Final quality inspection is a dynamic process. Based on the checkup stance capacity and test duration, it is often not possible to perform a thorough analysis for every product made. Such an inspection would be impossible from a time and financial point of view. Additionally, if the production process is well established and errors occur only rarely, it will not be necessary to conduct such an inspection on each piece produced. Random selection is often the first choice if some of the products shall be excluded from costly inspection. Targeting products with a higher probability of deviations, however, would result in more precise selection, saving expenditures from inspecting flawless products and decreasing the average production time. After passing the entire production process, the finished car is subject to a thorough analysis and tests, during which a group of experienced specialists determines whether the car meets rigorous quality standards. If any anomalies are found, the defective part of the car, whether it is an individual component or a series of them, must be reworked. The order, number, and depth of these inspections are set by each plant differently depending on the quality assurance strategy.

(1)Use Case 1: The road test is conducted at the end of the final inspection series as the very last test. Only a certain percentage of the vehicles produced are sent to the road test. The cars are selected randomly, which reduces the accuracy of targeting only the cars presumed to have defects. Combined with the fact that the testing process requires 20–30 min per car and personal capacities, the process is currently ineffective, and only 1–3% of cars are faulty in the rod test. Therefore, the selection strategy needs to be optimized. In this case, using AI is recommended as it is very effective for detecting unknown correlations among various input parameters and, in doing so, detects disturbance features very early. This allows action to be taken at an early stage—before the final product is sent to the customer—and economic damage to be limited. The aim is to increase the testing results by at least 400%, as per the market requirement. This means that, ultimately, fewer cars with failures will reach the final customer.(2)Use Case 2: The road test is conducted as the third test within the final inspection series. All vehicles produced are tested, even though the number of errors found is relatively small. This test is thus economically and personally challenging, as the duration of the RT lasts between 20 and 30 min, depending on the model. As proof of concept, the main aim is to lower the number of cars undertaking the RT by at least 10% and target the cars presumed to be flawless. Since the production cycle of one car takes 10–50 h and between 100,000 and 400,000 cars per year are produced, this would represent about 400,000 min per year saved on production time costs.

For both use cases, it is also necessary to note that to maintain high quality, the overall accuracy of the algorithm should reach at least 85%, as this is a specific requirement of the market.

## 3. Related Work

### Research Methodology

The focus of our research methodology is to develop a predictive quality inspection framework using advanced machine learning (ML) techniques. This framework aims to enhance quality control (QC) and quality management (QM) processes in automotive manufacturing, offering a sustainable, long-term solution.

Data Collection: Data are collected from various stages of the automotive manufacturing process, including production data, defect logs, and the final inspection results. This comprehensive data collection process ensures the models are trained on diverse scenarios and potential defect types.Data Preprocessing: The collected data are cleaned and normalized through preprocessing. This step involves handling missing values, normalizing numerical features, and encoding categorical variables. Effective preprocessing is crucial for ensuring the quality and reliability of ML models.Model Training: Multiple ML algorithms are employed to train the predictive models. The algorithms used include Decision Trees, Bayesian classifiers, K-Nearest Neighbors, Support Vector Machines, Logistic Regression, and Ensemble Methods. Each algorithm is evaluated for its performance in predicting defects, and the best-performing models are selected for deployment.Evaluation: The trained models are evaluated using metrics such as accuracy, precision, recall, and F1 score. Confusion matrices are used to assess the performance of the models in detecting defects. The models are also tested on historical data to ensure their robustness and reliability.Deployment: The best-performing models are deployed in the production environment. This involves integrating the models with existing QC and QM systems to provide real-time predictions and insights. The deployment process includes setting up infrastructure for data ingestion, model inference, and result visualization.Continuous Improvement: The deployed models are continuously monitored and retrained on new data to ensure their effectiveness over time. This involves setting up automated pipelines for data collection, preprocessing, model training, and deployment. Regular reassessments and model updates are crucial for adapting to changes in the manufacturing process and improving predictive accuracy.

Previous studies have explored various methodologies for enhancing QC and QM processes. Traditional methods, such as Six Sigma and Lean Manufacturing, focus on reducing variability and improving process efficiency [1,2]. However, these methods often lack real-time data analysis and AI capabilities.

Several studies have investigated the application of ML and AI in QC and QM. For example, ref. [8] discusses using artificial neural networks [9] (ANNs) for predicting product quality in plastic molding processes. Integrating IoT and Big Data into QM has also been shown to enhance decision-making and process efficiency [5]. However, these studies primarily focus on specific use cases and lack a comprehensive framework for continuous improvement.

Our research addresses these limitations by proposing a holistic and agile approach to predictive quality inspection. By integrating advanced ML algorithms and continuously updating the models based on new data, the proposed framework aims to provide a sustainable and effective solution for automotive manufacturing.

## 4. Materials and Methods

Based on extensive research and experience in processing Big Data and delivering predictive maintenance solutions, we amended and extended existing data mining methodologies into a framework for proposing and implementing a predictive strategy for quality inspection. We adapted the renowned and universally acknowledged CrossIndustry Standard Process for Data Mining (CRISP-DM) [4] to reflect the issue of continuous improvement in production and to ensure a durable solution for a customer. The CRISP-DM is a process model stemming from 1996 that was developed by the CRISP-DM consortium, consisting of DaimlerChrysler (later Daimler-Benz) Stuttgart, Germany, SPSS (later ISL) Chicago, IL, USA, NCR Systems Engineering Copenhagen, Denmark, and OHRA Verzekeringen en Bank Groep B.V. Arnhem, The Nederlands, with the aim of providing a universal strategy enabling the implementation of complex data mining projects. The framework includes the following crucial steps (as depicted in Figure 1):A: Business Understanding;B: Data Understanding;C: Data Preparation;D: Model Training;E: Evaluation;F: Deployment;G: Regular reassessment.

Improving production based on a regularly conducted analysis is characteristic of the manufacturing process in Industry 4.0. As there is a huge amount of data available to company analysts, production planners learn from previous defects, and production is constantly improving and evolving. Improvements in the manufacturing process and evolution in defects represented major challenges when developing the proposed strategy.

Therefore, in contrast to previous implementations such as those in [5], we proposed and implemented predictive quality inspection as a circular process, repeating the individual steps and following the business strategy goals throughout the entire process, not only at the beginning. An agile approach to solving the individual steps is crucial, as the development of both digital technologies and QM strategies is immense. The mere monitoring and maintenance of the deployed solution cannot be sustained in the long term.

Relying only on monitoring and maintenance is not sustainable, and omitting planning and implementing post-deployment strategies often leads to project termination after the proof-of-concept phase.

Hence, regular model recalibration and retraining are prerequisites for a sustainable solution. Furthermore, a constant reassessment of business strategy goals and the long-term suitability of the deployed solution is of central importance for the whole process. The standard data analysis methodology is implemented within the framework, which we also described in [10].

### 4.1. Business Understanding

Properly determining the objectives and requirements can save time and money in future steps. Even if the same issue is being solved, different requirements may occur based on the customer’s expectations or available sources.

Each type of manufacturing process, with specific process attributes, should be individually assessed to determine the suitability of using individual algorithms. Among others, the following goals will be taken into consideration:Increase the quality of production and the safety of the car for the end customer;Save production costs;Increase the availability and efficiency of production lines;Optimize the selected production parameters;Save production resources and materials;Save energy, CO_2_, water, etc., in production.

### 4.2. Data Understanding

Inspecting and obtaining relevant **data is** the first and most important step in the whole data mining process. A thorough analysis of the whole process shall be conducted within the plant in cooperation with the customer, and crucial data sources for the future predictive model shall be proposed. Potential problems with the data will be identified.

### 4.3. Data Preparation

Data obtained from various data sources should then be regularly adjusted and bulked to a data lake in agreed time slots. Data preprocessing includes data selection, cleansing, format processing, and/or data construction.

The data lake shall be selected based on different criteria, including costs, reaction time, and the type and amount of data. If there is a data lake already available within the business infrastructure, it is often an advantageous choice to add other data sources.

### 4.4. Model Training

In this section, we evaluate the methodological advantages of various binary classification methods used in our predictive quality inspection framework. Each method offers unique strengths and is suitable for different aspects of our application. By comparing these methods, we aim to highlight their specific benefits and demonstrate their effectiveness in predicting the quality of automobiles at the end of the production line. This comparison provides a clear rationale for selecting appropriate models based on the characteristics of the data and the requirements of the prediction tasks.

The methods evaluated include XGBoost, Linear Regression, Logistic Regression, K-Nearest Neighbors (KNN), Naive Bayes, Decision Tree, Support Vector Machine (SVM), and CatBoost. Below, we discuss each method’s advantages, supported by relevant sources.

**XGBoost** (Extreme Gradient Boosting)

XGBoost is a highly efficient and powerful implementation of gradient boosting. Its speed and performance are achieved through parallel processing and hardware optimization, making it suitable for large datasets and complex models. XGBoost has a built-in mechanism to handle missing data effectively, saving significant preprocessing time. It includes L1 and L2 regularization techniques, which help prevent overfitting by penalizing complex models. Additionally, XGBoost provides a way to measure the importance of each feature, aiding in feature selection and model interpretation. Its flexibility allows it to be applied to various types of data and problems, not just binary classification [11].


**Linear Regression**


Linear Regression is known for its simplicity and ease of implementation. It works best when the relationship between the independent and dependent variables is linear. Linear Regression is computationally efficient and fast, particularly for smaller datasets. This method is highly interpretable, as the coefficients directly indicate the relationship between each feature and the outcome, making it easy to understand and communicate the results. However, its application in binary classification is limited as it is fundamentally a regression technique [12,13].


**Logistic Regression**


Logistic Regression is widely used for binary classification due to its ability to provide probabilistic outputs for class membership. This can be particularly useful for decision-making processes where understanding the probability of an outcome is important. The coefficients in Logistic Regression are easy to interpret in terms of odds ratios, which provide insights into the influence of each feature on the outcome. Logistic Regression is computationally efficient and simple to implement. Moreover, it supports regularization techniques like L1 and L2, which help prevent overfitting and improve generalization [14,15].


**K-Nearest Neighbors (KNN)**


K-Nearest Neighbors (KNN) is an instance-based learning algorithm that is simple and easy to implement. It makes no assumptions about the data distribution, making it versatile for various types of data. KNN can capture complex patterns in the data through its non-parametric nature. It is also intuitive, as the classification decision is based on the majority class among the nearest neighbors. However, KNN can be computationally expensive for large datasets as it requires storing and searching through all training examples [16,17].


**Naive Bayes**


Naive Bayes classifiers are simple, fast, and effective for many binary classification problems. They are based on Bayes’ theorem and assume conditional independence between features, which often works well in practice despite being a strong assumption. Naive Bayes is particularly useful for high-dimensional datasets and performs well, even with limited training data. It also provides probabilistic outputs, which can be useful for decision-making processes [18,19].


**Decision Tree**


Decision Trees are highly interpretable models that split the data into subsets based on feature values. They can capture non-linear relationships and interactions between features without requiring any assumptions about the data distribution. Decision Trees are easy to visualize, making them useful for understanding and communicating the decision-making process. They can also handle both numerical and categorical data [20,21].


**Support Vector Machine (SVM)**


Support Vector Machines are powerful classifiers that work well for both linear and non-linear classification problems. They use kernel functions to transform the data into a higher-dimensional space, where it is easier to separate the classes with a hyperplane. SVMs are effective in high-dimensional spaces and are robust to overfitting, particularly in cases where the number of dimensions exceeds the number of samples. They also provide flexibility through the choice of different kernel functions [22,23].


**CatBoost**


CatBoost is a gradient-boosting algorithm specifically designed to handle categorical features without extensive preprocessing. It automates the handling of categorical variables, reducing the need for manual feature engineering. CatBoost also includes techniques to combat overfitting, such as ordered boosting, which ensures that the model’s predictions are less biased. It is efficient and scalable, making it suitable for large datasets and complex models [24].

After the data are analyzed and prepared in the data lake, model training can be conducted. Based on the data analysis, predicting the final car quality was assessed as a binary classification problem (OK/NOK state). A binary classification solution typically includes applying Logistic Regression, K-Nearest Neighbors, Decision Trees, Support Vector Machines, or Naive Bayes algorithms. If a larger number of features are included, boosting algorithms [25] with masking might also bring competitive results.

Below, we present examples of the use of algorithms for binary classification.

The issue of error detection in industrial production is addressed in [26]. The paper compares the use of Linear Regression and the XGBoost algorithm. The XGBoost algorithm turned out to be superior, achieving a better score. According to the author, Linear Regression, on the other hand, can reveal the impact of factors on the detection of faults in production parts.

In another paper, the authors focus on the use of Logistic Regression in the field of “Credit Card Fraud Detection” [27]. They achieved a very high accuracy value of 0.99. The administrator can detect fraud within the dataset of bank transactions through the software application. A weakness, as admitted by the authors themselves, is in the detection of online fraud. This will be the subject of further research.

Sometimes, a combination of algorithms helps in failure detection, as is the case in [28]. This involves the detection of motor bearing failures using feature extraction based on Spectral Kurtosis coupled with K-Nearest Neighbor Distance Analysis.

In [29], the authors concentrate on the task of detecting epileptic seizures through the analysis of EEG signals both in healthy subjects and those diagnosed with epilepsy. The detection methodology is predicated on the discrete wavelet transform (DWT) of EEG signals, employing both linear and non-linear classifiers. Specifically, the detection was executed by applying the Naive Bayes (NB) and K-Nearest Neighbor (K-NN) classifiers to statistical features derived from DWT. The findings reveal that the NB classifier demonstrated superior accuracy and efficiency in computational time in nine distinct datasets. In contrast, in four datasets, the K-NN classifier yielded enhanced precision but necessitated increased computational time.

In the current scientific landscape, an interesting use case is outlined in [30], in which a classification approach to predicting beef knuckle quality was researched using the Decision Tree and Naive Bayes methods. In this study, the authors describe the prediction of quality based on attributes such as the cooking water temperature, order time, second immersion, water volume, third immersion water volume, and amount of salt used. An important attribute included in the algorithm is not only the cooking process but also the subsequent drying process, i.e., the sequence and duration of each step. In addition, the weights of the animals, imperfections in the meat, qualitative parameters for meat evaluation, quality of transport services, etc., are taken into account. For the prediction, the classification algorithms C5.0 for Decision Tree and Naive Bayes were compared within the application system RapidMiner version 9.1. The Decision Tree showed an accuracy of 70%, while Naive Bayes showed an accuracy of 82%. This work underlines the importance of comparing different types of algorithms.

Another notable instance of binary classification is the statistical technique Support Vector Machine (SVM). References [31,32] present research on a Support Vector Machine-based assessment system on shift quality for vehicles. The study focuses on forecasting the standard of an automobile’s automatic transmission when it undergoes its final test drive toward the end of production.

The authors utilized an objective analysis, comparing sound signals by drivers with evaluations from sound sensors. The input data included sounds produced during automatic gear shifting, fuel consumption, and comfort ride evaluation. The model was trained on one type of car but has the ability to be applied to other types, thus predicting the quality of gear shifts. They utilized generalization in the SVM to achieve the results. The authors addressed a similar issue of predictive quality as the one we describe in this paper with the XGBoost algorithm in [33]. Multistage Quality Control is to be improved by applying machine learning to the automotive industry. Within the manufacturing process, the qualitative and safety indicators of the product (car) are monitored throughout production, if possible, but primarily at its conclusion. Sometimes, it is no longer feasible to repair the product, leading to its disposal. Alternatively, repair becomes time-consuming and financially burdensome. Although quality indicators within individual production steps show no defects and are evaluated as OK, at the end of the production cycle, they are evaluated as NOK. As mentioned by the authors in the cited work, during the assembly of car body parts into a complete three-dimensional shape, the perceptron evaluated geometric indicators such as NOK. Through a correlation analysis of the process and subsequent identification of an appropriate algorithm, they were able to predict error states. Similarly, in our research project described in this article, the Boost algorithms showed the best attribute values for accuracy and precision in the confusion matrix. We have had similar examples with our automotive production clients, such as compressors in the paint shop, where the interrelation of process attributes like pressure, revolutions, and current affects the final quality of the paint. Another example is the influence of torque sequences when tightening crucial structural bolts in cars. Neither humans nor devices (Perceptron for chassis alignment, Quality Eye Dog for paint quality identification, among others) can assess the influence a complex set of process attributes will have on the final quality and safety of a product. They can only judge from the perspective of localized measurement. However, with the mentioned algorithms, it becomes possible to automate this evaluation.

The introduction to CatBoost presented in [24] revealed the advantages of CatBoost as the implementation of ordered boosting, a permutation-driven alternative to the classic algorithm, and an innovative algorithm for processing categorical features, effectively solving the issue while reaching excellent empirical results. Recent publications reveal its effectiveness both in classification as well as regression tasks in the fields of economy and finance, meteorology, public health and medicine, psychology, marketing, geology, and other fields. The authors of [34] used CatBoost to predict loan defaults in peer-to-peer lending. The authors of [35] utilized it for recognizing daily life activities to promote healthier lifestyles and well-being. The authors of [36] aimed at predicting reference evapotranspiration in humid regions, while [37] used it to screen anxiety and depression among seafarers, and [38] used it for predicting online shopping behavior from click-stream data. The authors of [39] combined the CatBoost algorithm [40] with sequential model-based optimization for a real-time hard-rock tunnel prediction model for rock mass classification in geology.

In the context of this study, machine learning (ML) methods have been utilized to address various challenges in quality inspection and binary classification. To provide a clearer understanding of the existing literature and its relevance to our research, the references have been categorized into two distinct groups. The first group comprises studies that specifically focus on applying ML techniques within the domain of quality inspection, demonstrating the efficacy of these methods in detecting defects and improving production processes. The second group includes references that, while not directly related to quality inspection, explore the application of ML algorithms in other fields, particularly those involving binary classification problems. This distinction allows us to highlight the broader applicability of these methods as well as their specific impact on quality control in manufacturing.

Group 1—Using ML Methods in Quality Inspection:

Reference [26]: Discusses error detection in industrial production using Linear Regression and XGBoost algorithms, with XGBoost achieving superior results.Reference [28]: Addresses motor bearing failure detection using feature extraction based on Spectral Kurtosis combined with K-Nearest Neighbor Distance Analysis.Reference [33]: Describes the improvement of multistage quality control in the automotive industry through machine learning.References [31,32]: Research based on Support Vector Machines (SVMs) for assessing shift quality in vehicles.Reference [30]: A case study on predicting beef knuckle quality using Decision Tree and Naive Bayes methods.

Group 2—Using ML Methods for Other Purposes Regarding Binary Classification:

Reference [25]: Discusses the use of boosting algorithms with masking, which can yield competitive results when a large number of features are included in binary classification.Reference [29]: Focuses on detecting epileptic seizures through EEG signal analysis using Naive Bayes and K-Nearest Neighbor classifiers.Reference [24]: Introduces the advantages of the CatBoost algorithm, including its effectiveness in both classification and regression tasks.References [34,35,36,37,38,39]: Describe the application of CatBoost across various fields, including predicting loan defaults, recognizing daily life activities, forecasting evapotranspiration, screening anxiety and depression, predicting online shopping behavior, and forecasting geological properties.Reference [27]: Explores the use of Logistic Regression in the field of credit card fraud detection.

### 4.5. Evaluation

Trained models should always be double-checked against the current business strategy objectives. Obviously, before the model is to be deployed, overfitted models shall be excluded from the final decision-making step. It is proposed that the behavior of the model should be studied to understand how the prediction will change. If we apply a regression model and plot the predictions as a graph, as shown in Figure 2, then the peaks of this graph will be the predictions made by the model. The graph shows how each prediction is positioned relative to zero and one. In other words, it shows how confident the model is that the prediction belongs to one group or another. If the prediction is 1, then, from the point of view of the model, we can interpret this as close to a 100% probability that it is really 1. The same goes for predictions, the peaks of which are equal to 0 on the graph. From the model’s point of view, everything else has a probability between 0 and 1. Here, it becomes possible to independently regulate the level of probability, for which it can be assumed that the prediction is 0 or 1, thus regulating precision and recall.

### 4.6. Deployment

The model best fitting the set business strategy goals shall be implemented into the application programmed to fulfill the users’ needs. It does not include the machine learning model deployment alone. Instead, the development and deployment of the whole solution must be planned and conducted, including the following:Automatically obtaining and storing data, which become important for gradual learning and online result presentation;Implementing the prediction model and connecting it to the presentation layer;Developing the presentation layer of the final application so that it fulfills the requirement specification and, in addition to the AI model, the strict rules defined by the experienced quality inspectors might be amended;Deploying the ML layer;Deploying the presentation layer;Planning, developing, and deploying monitoring and maintenance in the form of a regular report or analytical evaluation dashboard;Planning and deploying a strategy for regular algorithm retraining and reassessment.

### 4.7. Regular Reassessment

Constant production improvement can present further challenges when introducing predictive approaches to maintenance or quality processes, as the problems faced last summer might not be present this year. The standard 12-month data training set and the application of the Champion model do not apply in this case. Regular algorithm reassessment and recalibration are musts to constantly improve the manufacturing process. Moreover, regularly updating the business strategy and defining new goals are often necessary in the rapidly evolving business and production environment of a factory. Hence, the self-learning capability of the algorithm needs to be assured since the whole production process is fine-tuned and avoids repeating old errors. At the same time, new errors and problems are created, sometimes by changes at the production line or by changes in individual vehicle parts because of issues at the supplier side.

## 5. Results

We implemented the presented framework into predictive quality inspection during the road test at two different automotive plants. Although there are different processes in car manufacturing and different goals were set for their implementation, the proposed framework could have been utilized in both factories, bringing automation to quality inspection by predicting car defects during road tests.


**Business understanding**


UC1: A road test represents the very last test within the quality inspection. It is oriented toward checking undesirable sounds in a car. The road test was appended. The aim was to determine the final quality of the car at the road test as either OK or NOK with the highest possible accuracy. The cars predicted to be NOK should have been recommended for the road test. The number of tested cars was expected to reach approximately 10% of the total production.UC2: Reducing the number of cars sent to the road test was identified as the main goal for the second use case, starting with diminishing the total count of tested cars by 10%. The selection process was supposed to be conducted in a way that preferred omitting cars with the highest probability of being OK in the road test. The number of tested cars will be approximately 90% of the total production.


**Understanding and preprocessing the data**


To facilitate the selection of an optimal data mining strategy, a comprehensive understanding of the data is paramount. The following data sources were used for the primary analysis:Audit data from the body shop, paint shop, and assembly process;Selected process data from production;Defects were identified at road tests.

The foundation of our analysis rests on a dataset encompassing the historical quality of cars.

By thoroughly examining the offline data acquired over a period of 12 months, we embarked on a preliminary analysis to determine the pertinent information for data mining purposes. We meticulously scrutinized the format, quantity, and occurrence of the data, unraveling the intricate relationships among its various attributes while evaluating its quality and cleanliness.

UC1: In this case, the data were presented in tabular form with 165,988 rows and 19 columns. An analysis of the data revealed a number of problems/shortcomings that were eliminated before they were transferred to the model. One of these shortcomings was the classic duplication of some strings (this error most likely occurred when writing the database). Furthermore, although each column was unique, some of them had duplicate information. The only difference was in the form in which this information was recorded. Thus, these columns were only redundant. Some of the columns by themselves did not carry any useful information to solve the problem; these were also eliminated from the data.UC2: The dataset was organized in tabular form, comprising a frame size of 399,133 rows and 19 columns. Upon conducting our analysis, we unearthed several inconsistencies within the dataset. These discrepancies included the presence of multiple car models, despite the originally agreed intention to focus solely on a single model. We also discovered quality events that were captured after the road test, rendering their inclusion in the prediction process inappropriate. Additionally, certain event types featured duplicate information with disparate identifiers, and some cars exhibited multiple results from the road test. Furthermore, we encountered instances of missing values for one of the attributes, further complicating the integrity of the data.

The dataset preprocessing phase encompassed a series of sophisticated steps, ensuring its suitability for subsequent analysis. These steps encompassed data conversion, parsing when necessary, eliminating both quantitative and logical duplicates, and evaluating the target variable, necessitating its transformation into a format amenable to the model’s requirements.

The data underwent a meticulous cleansing process involving the removal of redundant features from the original set. Attributes were identical across all cars, and those directly identifying a car were removed. Furthermore, pseudo-errors of the “no fault found” type were eliminated from the training process as they did not meaningfully contribute to the analysis. Following this, a type conversion from Integer to Categorical Input was performed for several attributes, aligning the feature representations with the nature of the information they conveyed. To appropriately handle the nominative error variable, we applied the OneHotEncoder approach. To augment the initially acquired datasets, we introduced additional attributes derived from a combination of existing features, the occurrence numbers of specific characteristics, and temporal units such as months or calendar weeks.


**Data integration**


Data utilized for the present predictive quality inspection were collected from original relational databases and transformed and bulked into a data lake, which was subsequently used as the data source for both model training and online predictions. Based on the previously conducted offline analysis, we identified meaningful data sources. Most of the data were obtained utilizing a pipeline system. Conducting data preprocessing while obtaining data enables us to fill the data lake with data already being processed and ready to be analyzed. To avoid overloading the source database, all pipelines work with incremental data only. Data that are not available from source databases in real-time were sent to the data lake directly via TCP/IP and MQTT protocols from the production systems, as presented in the actual data flow depicted in Figure 3.


**Model training and evaluation**


Prior to training individual models, we established a comprehensive test design to ensure the accurate evaluation of the models. Our objective was not only to assess the performance of the data during the training and testing periods but also to obtain reliable models through rigorous validation and cross-validation procedures. To achieve this, we partitioned the data into distinct sets for training, testing, and validation. Specifically, we adopted the following test design:Training set: 70% of the data;Test set: 30% of the data;Validation set: Data from a time period not covered by the previous sets, ideally encompassing the last 2–3 months of the most recent 30%.

As an initial and intuitive approach for binary classification, we opted to train commonly used models, including a Decision Tree classifier, Random Forest classifier, Support Vector Classifier (SVC), Logistic Regression, K-Nearest Neighbors classifier, XGBoost, and Catboost. Both ensemble algorithms and regressors present viable solutions for failure prediction. However, the appropriate configuration, preprocessing of input data, and validation of results across different time periods are crucial for obtaining reliable models.

UC1: Several different vehicle types with the same platform were part of this use case. Approximately 10% of the produced cars are to undergo the road test. The aim of the project was to target faulty cars and test them. Table 1 presents the results of predicting different models on raw data. The data underwent only primary processing, such as that described above. No feature engineering was applied. Classification algorithms were also not configured but used with default settings. In Table 2, the results of the algorithms with default settings are presented but with modified data. The usual OneHotEncoding was used for several key features.

CatBoost provided the best results in the training/testing sets, reaching the following metrics: accuracy, 0.9379; precision, 0.9083; and recall, 0.589. Validation was carried out on new data that the algorithms had not yet processed. According to the validation results, the algorithm allowed for the number of checked cars to be reduced from 1530 to 202, and 99.09% of the cars were predicted to be OK as they had not demonstrated any errors. The selected CatBoost strategy promised 59% of correctly predicted NOK cars. The proposed solution was accepted by the customer and deployed. The validation results are displayed in Table 3. After deployment, the Champion model reached approximately 30% NOK cars, which still represents a major increase compared to the ca. 5% standard NOK rate previously reached with random selection.

UC2: The aim was to reduce the number of cars undergoing the quality test while detecting as many NOK cars as possible. Two different car models were to be predicted: an established model and a newly introduced model. We trained different models on raw data, i.e., without OneHotEncoder. The overall results of the work of various algorithms for pure, unmodified data are displayed in Table 4. In this case, the models are not fine-tuned; they are used “out of the box.”

To generate additional features, we suggested using OneHotEncoder for the nominative features of the most important categorical attributes. Ensemble algorithms work best in this situation. The results are visualized in Table 5. Vehicle Type 1 is an established car model, with production already tuned up. According to the training/testing sets, using the Champion model would enable us to reduce the number of cars undergoing the quality test from 3325 to 419 while aiming for 91.5% of NOK cars. To validate the results, data from an independent time period were used. The validation results indicate that targeting NOK cars was reduced to 84.62%.

Vehicle 2 was a new vehicle with a ramp-up phase, a high error rate, and no previous experience with the car. The decision was easier as the ratio of NOK cars was relatively high. Validation after several algorithm adjustments and after the start phase revealed that the number of NOK cars is considerably lower, but the algorithm still maintains its NOK target at 87.77%. These numbers were accepted by the customer, and the models were deployed for production. The confusion matrices for the validation sets are depicted in Table 6.


**Model deployment**


Both use cases were deployed within the business infrastructure, fulfilling the security requirements and providing working products. All parts of the project were implemented into the existing intranet with no access to cloud solutions. All technology was selected based on the provided budget and available data sources. The data lake was implemented by utilizing Elastic Stack components. The NoSQL record-based database Elasticsearch was utilized as data storage for both online and offline data. Data from RDBMS were obtained, transformed, and loaded into the database using the Logstash component. Monitoring was set and provided utilizing Metricbeat and Heartbeat.

The visual layer of the solution consists of the main dashboard depicting traffic light colors for cars, representing the decision of the algorithm, and two QM-supporting dashboards. For UC1, decision-supporting rules were implemented, ruling out cars for which it was impossible to undergo the test, e.g., because of more extensive packaging. For UC2, not only was the dashboard prepared to visualize the result of the road test, but the prediction results were also sent to an actual traffic light installed directly in the production hall. The decision dashboard was supported by two evaluation pages for quality management, enabling the results of the algorithm and the road test to be reported and analyzed as such. The data depicted in the reporting parts of the application were selected by the QM experts. Furthermore, monitoring was set up for the whole solution. The following notifications were sent:An infrastructure problem (Elasticsearch/Logstash/Beats);An error state in the Web Application logs;An error message in the predictor logs;An error in the relearning logs;A better algorithm is prepared and automatically deployed.
**Regular algorithm revision**

Currently, production in the automotive industry invests a considerable amount of resources into analyzing and resolving known issues. If there are any reoccurring defects in the production process, the quality department strives to identify the cause and resolve the problem in coordination with analysis centers. As a reaction to constantly evolving and improving production, a system for automated retraining, recalibration, and algorithm reassessment was developed and delivered with the proposed solution. The self-learning application provides algorithms for training, comparing, and deploying new AI models into a predictor, which uses the models to make online prediction calculations. A flow diagram of the ready-to-use system is depicted in Figure 4.

The evaluation is conducted based on agreed evaluation criteria derived from the business strategy goals with the possibility of updating or changing the training process only by delivering a new “evaluation function”.

Tree-based algorithms are generally not sensitive to data normalization; in cases where there is a significant difference in the range of different features, the results for normalized data may differ from those for non-normalized data. This may affect the choice of the splitting thresholds, which is a feature observed in some of our data.

In scenarios where there is a significant difference in the range of features, normalization can indeed affect the performance of the model, although the impact is typically small compared to other types of algorithms, such as SVMs or neural networks.

The main reason why normalization can affect tree-based models is due to the way the splitting thresholds are determined. If one feature has a much larger range of values compared to others, it could disproportionately influence the decision-making process at each split in the tree. This is because the larger range could cause the model to select splits that are optimal for the feature with the larger range, potentially ignoring smaller but significant variations in other features.

In addition, some empirical evidence suggests that in certain contexts, especially where numerical precision or floating-point handling is critical, normalization could slightly alter the structure of the tree. This could lead to different splits being chosen, thus affecting the predictions of the model or its generalization capabilities.

This paper [41] discusses the impact of data normalization on the performance of machine learning algorithms, in particular decision tree-based algorithms. It was experimentally found that the proposed normalization method increases the accuracy of decision tree-based models (such as the Decision Tree and Extra Trees Classifier) by 1–6%, depending on the classification task (binary or multi-class).

## 6. Discussion

The results from our study demonstrate the efficacy of the proposed predictive quality inspection framework, which integrates AI capabilities to improve automotive manufacturing processes. By leveraging machine learning techniques, our framework successfully predicts the quality of automobiles at the end of the production line, leading to enhanced defect detection and personalized road tests. This approach significantly improves production efficiency and reduces costs.

However, there are several aspects that warrant further discussion. One critical area is the integration of additional data sources. While our current implementation shows promising results, expanding the scope to include more diverse data from different stages of the manufacturing process could further enhance the accuracy and reliability of the predictions. Additionally, incorporating feedback from service centers, where customers report technical issues during the warranty period, could provide valuable insights into long-term vehicle performance and common defects.

Another important consideration is the role of Generative AI technologies. By classifying defects from all service centers, Generative AI can help identify patterns and predict potential issues before they become significant problems. This proactive approach not only improves vehicle quality but also enhances customer satisfaction by reducing the likelihood of recurring defects.

Finally, the development of universal predictive maintenance applications is crucial for further advancing manufacturing quality and efficiency. By creating adaptable and scalable solutions, manufacturers can apply these predictive models across various production lines and facilities, ensuring consistent quality control and continuous improvement.

## 7. Conclusions

In this study, we present an innovative software framework with AI capabilities for predicting the quality of automobiles at the end of the production line. Our framework significantly improves defect detection and supports personalized road tests, enhancing production efficiency and reducing costs. The implementation results demonstrate that integrating AI into quality control processes provides a sustainable, long-term solution for continuous improvement in automotive manufacturing. The economic benefit is substantial, with potential savings of hundreds of thousands of production minutes annually.

Future work will focus on expanding production line process data sources. Additionally, Generative AI technologies will be incorporated to assist in classifying defects from all service centers at which customers report technical issues during the warranty period.

## Figures and Tables

**Figure 1 sensors-24-05644-f001:**
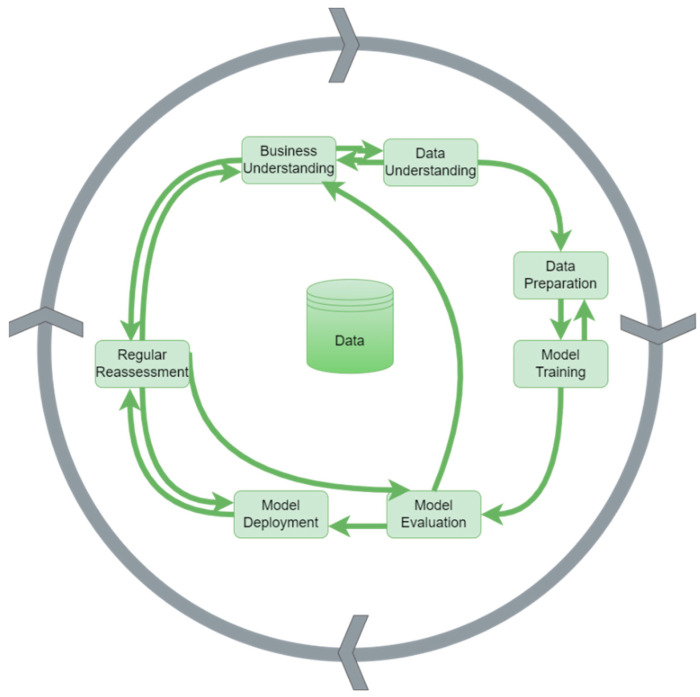
Predictive quality inspection: CRISP-DM adaptation.

**Figure 2 sensors-24-05644-f002:**
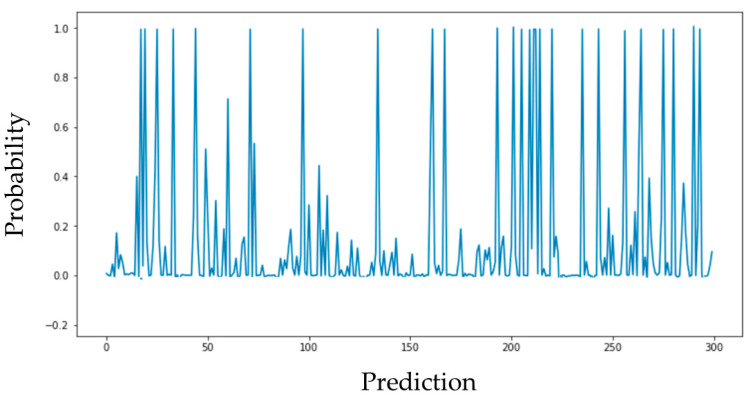
Predictions are plotted as a graph.

**Figure 3 sensors-24-05644-f003:**
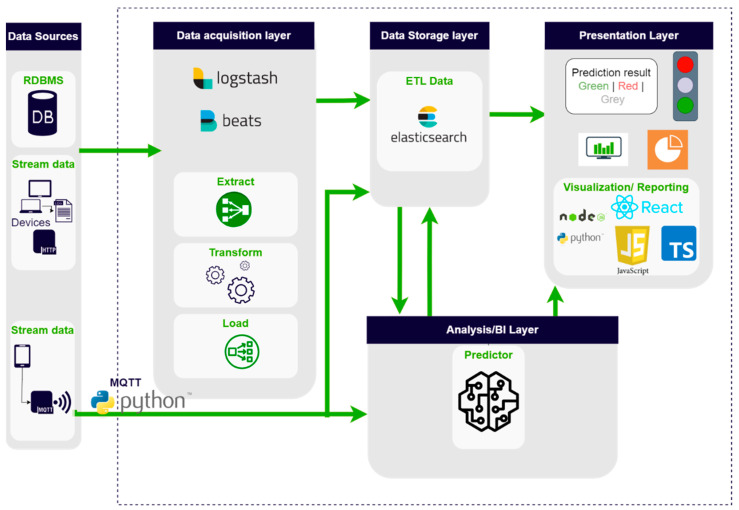
Data flow system: actual implementation from data integration to visual layer.

**Figure 4 sensors-24-05644-f004:**
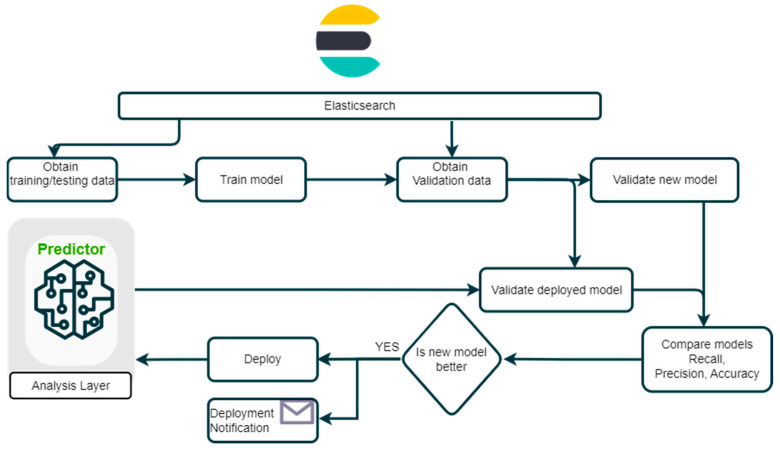
Flow diagram: system for automated retraining, recalibration, and algorithm reassessment.

**Table 1 sensors-24-05644-t001:** UC1: Results of predicting different models on raw data.

Classifier	Accuracy	Precision	Recall
Decision Tree Classifier	0.79	0.23	0.23
Random Forest Classifier	0.86	0.28	0.03
SVC	0.87	0.11	0.00
Logistic Regression	0.87	0.00	0.00
KNeighbors Classifier	0.85	0.16	0.03
XGBoost	0.85	0.24	0.06
CatBoost	0.86	0.30	0.02

**Table 2 sensors-24-05644-t002:** UC1: Results of predicting different models on modified data.

Classifier	Accuracy	Precision	Recall
Decision Tree Classifier	0.92	0.71	0.62
Random Forest Classifier	0.91	0.90	0.32
SVC	0.87	0.00	0.00
Logistic Regression	0.87	0.00	0.00
KNeighbors Classifier	0.85	0.16	0.03
XGBoost	0.94	0.88	0.59
CatBoost	0.94	0.91	0.59

**Table 3 sensors-24-05644-t003:** UC1: Validation results for the time-independent dataset.

Classifier	OK	NOK	Value Relations, Percent
Pred OK (CatBoost)	1316	12	99.09%
Pred NOK	83	119	58.91%
Value relations, percent	94.07%	90.83%	
Pred OK (Tree)	1277	51	96.16%
Pred NOK	77	125	61.88%
Value relations, percent	94.31%	71.02%	
Pred OK (RandomF)	1321	7	99.47%
Pred NOK	138	64	31.68%
Percent	97.2%	90.14%	
Pred OK (KNeigh)	1296	32	97.59%
Pred NOK	196	6	2.97%
Value relations, percent	86.86%	15.78%	
Pred OK (XGBoost)	1312	16	98.79%
Pred NOK	83	119	58.91%
Value relations, percent	94.05%	88.15%	

**Table 4 sensors-24-05644-t004:** UC2: Results of predicting different models on raw data.

Classifier	Accuracy	Precision	Recall
Decision Tree Classifier	0.77	0.21	0.21
Random Forest Classifier	0.85	0.31	0.03
SVC	0.85	0.00	0.00
Logistic Regression	0.84	0.00	0.00
KNeighbors Classifier	0.84	0.19	0.03
XGBoost	0.85	0.41	0.06
CatBoost	0.86	0.56	0.02

**Table 5 sensors-24-05644-t005:** UC2: Results of predicting different models on modified data.

Classifier	Accuracy	Precision	Recall
Decision Tree Classifier	0.93	0.75	0.76
Random Forest Classifier	0.96	0.98	0.71
SVC	0.86	0.00	0.00
Logistic Regression	0.86	0.00	0.00
KNeighbors Classifier	0.84	0.19	0.03
XGBoost	0.95	0.95	0.77
CatBoost	0.96	0.99	0.71

**Table 6 sensors-24-05644-t006:** UC2: Confusion matrices for the CatBoost algorithm applied to validation sets; data are displayed per vehicle type.

**Vehicle Type 1**	**OK**	**NOK**	**Value Relations, Percent**
CatBoost			
Pred OK	595	28	95.51%
Pred NOK	184	154	45.56%
Value relations, percent	76.38%	84.62%	
**Vehicle Type 2**	**OK**	**NOK**	**Percent**
CatBoost			
Pred OK (Tree)	5234	46	99.13%
Pred NOK	354	330	48.25%
Value relations, percent	93.66%	87.77%	

## Data Availability

Data are contained within the article.

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
