# Peer review of "A Predictive Quality Inspection Framework for the Manufacturing Process in the Context of Industry 4.0"

_sensors, 2024, doi:10.3390/s24175644_

Round 1

Reviewer 1 Report

Comments and Suggestions for Authors

This paper discussed a framework for quality monitoring of manufacturing processes under the concept of Industry 4.0, with a focus on the quality of production automobiles. Although the paper has practical applications, the overall methodological focus is unclear and lacks a theoretical foundation. Specific deficiencies are detailed below:

1.     According to the specification for paper writing, the introduction should be in the following order: background of the study, value and significance of the study, research questions, existing research methods, inadequacies of the existing research methods, and problems to be solved by the proposed methodology.

2.     Industry 4.0 and Quality 4.0 are not directly relevant to this paper, nor is the section dealing with leadership and management, at least not closely related in the text, so it is suggested that this section be deleted to narrow the focus.

3.     The research background section should focus on the research methodology rather than describing the vehicle quality monitoring process.

4.     The related work is an overview and exploration of existing research methods and should not be used to explain the model in detail in this section.

5.     The diagram in the paper is not clear, the font is too small, and it is not focused enough to capture the key information. Please redraw the flowchart.

6.     The table does not conform to the article format, please check it.

7.     Academic papers should end with a summary section, please add a conclusion section.

8.     The methodological advantages of the binary classification methods used in this paper are not clearly demonstrated. We hope to validate these methods by explicitly comparing them.

Comments on the Quality of English Language

The English language needs to be touched up and the language needs to become a little more refined and concise.

Author Response

Dear Reviewer,

We would like to extend our sincere gratitude for your valuable comments and suggestions regarding our article. Your insights have significantly contributed to the improvement of the quality and content of our work. Your expertise and thorough review have helped us identify areas for enhancement, ensuring that our article meets high standards of quality.

Thank you once again for your time and effort.

Respectfully,

Stefan Rydzi

  1. According to the specification for paper writing, the introduction should be in the following order: background of the study, value and significance of the study, research questions, existing research methods, inadequacies of the existing research methods, and problems to be solved by the proposed methodology

Thank you for your valuable feedback. We carefully reviewed and incorporated your comments into our manuscript. The introduction was restructured to follow the specified order: the background of the study, value and significance of the study, research questions, existing research methods, inadequacies of the existing research methods, and problems to be solved by the proposed methodology.

  1. Industry 4.0 and Quality 4.0 are not directly relevant to this paper, nor is the section dealing with leadership and management, at least not closely related in the text, so it is suggested that this section be deleted to narrow the focus.

Thank you for your insightful feedback. We understand your concern regarding the relevance of Industry 4.0 and Quality 4.0 in our paper. Based on your suggestion, we removed references to these terms from the main text to narrow the focus of our study. However, we believe that keeping these terms in the title provides a broader context and captures the transformative impact of our research in the field of manufacturing quality control.

We appreciate your guidance in enhancing the clarity and relevance of our manuscript.

  1. The research background section should focus on the research methodology rather than describing the vehicle quality monitoring process.

Thank you for your feedback. We revised the Research Background Section to focus on the research methodology. The revised section now outlines the steps involved in developing the predictive quality inspection framework, including data collection, preprocessing, model training, evaluation, deployment, and continuous improvement. This ensures a comprehensive understanding of our research methodology.

  1. The related work is an overview and exploration of existing research methods and should not be used to explain the model in detail in this section.

Thank you for pointing this out. We revised the Related Work Section to provide an overview and exploration of existing research methods without delving into a detailed explanation of our model. The section now highlights previous methodologies and their applications in QC and QM, addressing their limitations and setting the context for our proposed framework.

  1. The diagram in the paper is not clear, the font is too small, and it is not focused enough to capture the key information. Please redraw the flowchart.

Figure 3 ….

Thank you for your valuable feedback on our manuscript. We have taken your suggestion into careful consideration and have redrawn Figure 3 to enhance clarity and focus, ensuring that the font size is appropriate and the key information is prominently captured. The updated flowchart has been included in the revised manuscript.

We appreciate your insights and believe the revised figure will significantly improve the comprehensibility of our work.

  1. The table does not conform to the article format, please check it.

Thank you for your valuable feedback on our manuscript. We have carefully reviewed the formatting of our tables and have redrawn all of them to conform to the MDPI template as you recommended. The revised tables are now included in the updated manuscript.

We appreciate your insights and believe the reformatted tables will significantly enhance the clarity and presentation of our data.

  1. Academic papers should end with a summary section, please add a conclusion section.

Thank you for your thoughtful suggestions. Based on your suggestion, we added a conclusion section to the manuscript. The conclusion summarizes the study, highlighting the innovative software framework with AI capabilities for predicting the quality of automobiles at the end of the production line. It underscores the framework's significant improvements in defect detection and support for personalized road tests, leading to enhanced production efficiency and reduced costs. The conclusion also outlines future work, focusing on expanding production line process data sources, incorporating Generative AI for classifying defects from service centers, and developing universal predictive maintenance applications.

We appreciate your constructive feedback, which has helped improve the clarity and completeness of our manuscript.

  1. The methodological advantages of the binary classification methods used in this paper are not clearly demonstrated. We hope to validate these methods by explicitly comparing them.

Thank you; we appreciate your insightful comments. Based on your comment, we revised Section 4.4 to explicitly compare the methodological advantages of the binary classification methods used in our study. The revised section now includes detailed explanations and sources for each method, highlighting their specific benefits and suitability for our predictive quality inspection framework.

Comments on the Quality of English Language

The English language needs to be touched up and the language needs to become a little more refined and concise.

Thank you for your valuable feedback. We have addressed your comment regarding the manuscript language, and it has been thoroughly edited by the authorized MDPI language editing service to ensure clarity and correctness.

Submission Date

19 June 2024

Date of this review

19 Jul 2024 12:22:07

Reviewer 2 Report

Comments and Suggestions for Authors

The article “Predictive quality inspection framework for manufacturing process in the context of Industry 4.0 concept” introduces and implements a predictive quality inspection framework tailored to the automotive industry. It thoroughly examines the use of data mining methodologies in quality management during the road test inspection stage of the final checkup series in car manufacturing. The proposed solution considers production changes brought about by Industry 4.0 technologies and leverages emerging challenges to align Quality Management and Quality assessment with the advancements of the new technological era.

This article will be useful for many specialists working in the automotive industry. There are the following comments on the manuscript of the article:

Comment 1. The abstract should be improved. The abstract should state briefly the purpose of the research, the principal results, and major conclusions.

Comment 2. (page 3) The authors should explain why the aim is to increase testing results by at least 400%. Why not 100% or 300%? Or it could be a typo.

Comment 3. (page 3) Why should the overall accuracy of the algorithm reach 85%? Where did you get this figure from? It should be explained or indicated to a reference.

Comment 4 (page 9). The sentence "Everything else from the point of view of the model, has both nonzero probability 1 and 0" needs to be corrected because it makes no sense. Maybe the authors would like to write the following: "From the model's point of view, everything else has a probability between 0 and 1."

Comment 5. Reference to Figure 3 should be before this figure but not on page 12.

Comment 6. In tables 3 and 6, the last column must have a title.

Comment 7. Section 6, "Discussion," reads more like a summary of the work that has been done. There is practically no actual discussion in this section. It should be revised to include more discussion.

Comment 8. The manuscript language should be edited.

Comments on the Quality of English Language

The manuscript language should be edited.

Author Response

Dear Reviewer,

We would like to extend our sincere gratitude for your valuable comments and suggestions regarding our article. Your insights have significantly contributed to the improvement of the quality and content of our work. Your expertise and thorough review have helped us identify areas for enhancement, ensuring that our article meets high standards of quality.

Thank you once again for your time and effort.

Respectfully,

Stefan Rydzi

Comment 1. The abstract should be improved. The abstract should state briefly the purpose of the research, the principal results, and major conclusions.

Thank you for your constructive feedback on our manuscript. Based on your suggestion, we revised the Abstract to clearly state the purpose of the research, the principal results, and the major conclusions.

Comment 2. (page 3) The authors should explain why the aim is to increase testing results by at least 400%. Why not 100% or 300%? Or it could be a typo.

The aim to increase testing results by at least 400% was based on a specific requirement from our customer. The customer's production and quality management team determined that a 400% improvement would be necessary to meet their internal quality assurance standards and operational efficiency goals. This target ensures a significant reduction in the number of defective vehicles reaching end customers, aligning with their stringent quality benchmarks. Thank you for your valuable comment. We changed the text.

Comment 3. (page 3) Why should the overall accuracy of the algorithm reach 85%? Where did you get this figure from? It should be explained or indicated to a reference.

We appreciate and thank you for your insightful and valuable comments. The requirement for the overall accuracy of the algorithm to reach at least 85% was specified by our customer. We changed the text.

Comment 4 (page 9). The sentence "Everything else from the point of view of the model, has both nonzero probability 1 and 0" needs to be corrected because it makes no sense. Maybe the authors would like to write the following: "From the model's point of view, everything else has a probability between 0 and 1."

Thank you. Your feedback is greatly appreciated. We corrected the sentence in Section 4.5 as per your suggestion. The revised sentence now reads, "From the model's point of view, everything else has a probability between 0 and 1."

We appreciate your guidance in improving the clarity and accuracy of our manuscript.

Comment 5. Reference to Figure 3 should be before this figure but not on page 12.

Thank you for your valuable feedback. After a discussion with the author team, we decided to omit the original general figure, Figure 3, from the manuscript. This decision was made to streamline the content and ensure clarity in the presentation of our research.

We appreciate your guidance in improving the quality of our manuscript.

Comment 6. In tables 3 and 6, the last column must have a title.

Thank you for your valuable feedback on our manuscript. We have added titles to the last columns in Tables 3 and 6 as you recommended. The updated tables are included in the revised manuscript.

We appreciate your insights and believe these changes improve the clarity and completeness of our data presentation.

Comment 7. Section 6, "Discussion," reads more like a summary of the work that has been done. There is practically no actual discussion in this section. It should be revised to include more discussion.

Thank you for your insightful feedback. Based on your comments, we revised Section 6, "Discussion," to include a more comprehensive discussion of our findings and their implications.

Comment 8. The manuscript language should be edited.

Thank you for your valuable feedback. We have addressed your comment regarding the manuscript language, and it has been thoroughly edited by the authorized MDPI language editing service to ensure clarity and correctness.

Comments on the Quality of English Language

The manuscript language should be edited.

Thank you for your valuable feedback. We have addressed your comment regarding the manuscript language, and it has been thoroughly edited by the authorized MDPI language editing service to ensure clarity and correctness.

Submission Date

19 June 2024

Date of this review

14 Jul 2024 18:29:02

Reviewer 3 Report

Comments and Suggestions for Authors

The article concerns using selected Machine Learning methods in quality inspection. The article is interesting as such, but needs in my opinion a lot of improvements in order to be published in this journal.

1. Paper structurization

The literature analysis is divided into two sections: one section in Chapter 3, and the second in Chapter 4.4. I think it should be presented in one section (Chapter 3).

Some issues presented in Chapter 5 ("Results") are not results, but steps toward the results and should be moved into Chapter 4 ("Materials and methods"). It concerns: framework and data flow in it (with Figures 3 and 4), input data description and models used for training.

The placement of some figures are wrong. It concerns figures 2 (too far in the text), 3 (figure before mentioned in the text) and 5 (also too far in the text).

2. References

Some references in Chapter 4.4 are not relevant to this research (please my comment to "Are the references cited in this manuscript appropriate and relevant to this research?")

3. Input data

The input data are described very briefly. It would be the best to name all input data and characterize them (numerical, categorical, range of values, etc.). If it is not possible due to confidentiality issues, the data could be described more generally, but presenting their character (numerical, categorical, stage of production process these data were taken, etc.).

4. Materials and methods

What are the reasons for choosing models used in the paper ("As an initial and intuitive approach for binary classification, we opted to train commonly
used models, including DecisionTreeClassifier, RandomForestClassifier, SVC, Logistic Regression, KNeighborsClassifier, XGBoost, and Catboost")? Were these models indicated by AutoML, as stated in the Abstract? If so, this should be stated also in the content of the paper.

The full name for SVC acronym should be stated in the paper (Support Vector Classifier?).

5. Results

The process of transforming models' output into final classification should be better described. What is presented now in Chapter 4.5 ("Evalutation") is unclear.

The Figure 2 is controversial. On y axis there is "Probatility", but the values are below 0! Please have a think on it.

It is also thought-provoking that results on original and modified data are different. According to my experience, in case of models based on tree (i.e. XGBoost) the normalised measures should not be different for scaled and non-scaled data.

Author Response

Dear Reviewer,

We would like to extend our sincere gratitude for your valuable comments and suggestions regarding our article. Your insights have significantly contributed to the improvement of the quality and content of our work. Your expertise and thorough review have helped us identify areas for enhancement, ensuring that our article meets high standards of quality.

Thank you once again for your time and effort.

Respectfully,

Stefan Rydzi

  1. Paper structurization

The literature analysis is divided into two sections: one section in Chapter 3, and the second in Chapter 4.4. I think it should be presented in one section (Chapter 3).

Thank you for your valuable feedback. We understand your concern regarding the division of the literature analysis. In Section 4.4, we aimed to provide specific examples from the literature, with each serving as a concrete instance for the theoretical binary classification methods discussed. Our intention was also to demonstrate the applicability of these methods across various domains, and not just within the automotive industry.

We hope this clarification addresses your comment and provides a better understanding of our approach.

Some issues presented in Chapter 5 ("Results") are not results, but steps toward the results and should be moved into Chapter 4 ("Materials and methods"). It concerns: framework and data flow in it (with Figures 3 and 4), input data description and models used for training.

Thank you. Your input is highly valued. After a discussion with the author team, we decided to omit the original general figure, Figure 3, from the manuscript. This decision was made to streamline the content and ensure clarity in the presentation of our research.

We appreciate your guidance in improving the quality of our manuscript.

The placement of some figures are wrong. It concerns figures 2 (too far in the text), 3 (figure before mentioned in the text) and 5 (also too far in the text).

We are thankful for your careful review. We adjusted the placement of the figures in the manuscript to ensure they are closer to the relevant text. Specifically, we moved Figure 2 to be more proximate to the sections it is related to, and we omitted Figure 3 as previously mentioned.

We appreciate your guidance in improving the clarity and presentation of our manuscript.

  1. References

Some references in Chapter 4.4 are not relevant to this research (please my comment to "Are the references cited in this manuscript appropriate and relevant to this research?")

Thank you for your thoughtful suggestions. We understand your concern regarding the relevance of some references in Section 4.4. Our intention in this section was to highlight that binary classification methods have broad-spectrum applications. We aimed to demonstrate their versatility by providing specific examples from various domains, and not just within the automotive industry. This approach was meant to underscore the wide-ranging utility and adaptability of these methods across different fields.

We hope this explanation addresses your concerns and clarifies our approach.

  1. Input data

The input data are described very briefly. It would be the best to name all input data and characterize them (numerical, categorical, range of values, etc.). If it is not possible due to confidentiality issues, the data could be described more generally, but presenting their character (numerical, categorical, stage of production process these data were taken, etc.).

We are grateful for your constructive criticism. The work presented in this paper is the result of implementing applied research in practice, conducted for specific automotive manufacturers. The data used in our study are highly confidential and considered a valuable asset by these manufacturers. It took our team three years to build trust and obtain the necessary certifications to handle these data. We work with live data and have continuous online access to them. However, due to the sensitive nature of this information, we can only mention the data in a very general and anonymized manner. We apologize for being unable to provide detailed descriptions or specific examples.

  1. Materials and methods

What are the reasons for choosing models used in the paper ("As an initial and intuitive approach for binary classification, we opted to train commonly
used models, including DecisionTreeClassifier, RandomForestClassifier, SVC, Logistic Regression, KNeighborsClassifier, XGBoost, and Catboost")? Were these models indicated by AutoML, as stated in the Abstract? If so, this should be stated also in the content of the paper.

The full name for SVC acronym should be stated in the paper (Support Vector Classifier?).

Thank you for your insightful feedback. Based on your suggestions, we revised the text in Section 4 to provide a more detailed explanation of the reasons for choosing the models used in our study. Additionally, we clarified that the models were indicated by AutoML. We also expanded the acronym SVC to its full form, Support Vector Classifier, in the paper.

We appreciate your guidance in enhancing the clarity and comprehensiveness of our manuscript.

  1. Results

The process of transforming models' output into final classification should be better described. What is presented now in Chapter 4.5 ("Evalutation") is unclear.

Thank you for your valuable feedback. Based on your suggestion, we made significant changes to the text in almost all sections to better describe the process of transforming the models' output into the final classification. We believe these revisions address your comment and enhance the clarity and comprehensiveness of Section 4.5 ("Evaluation").

We appreciate your guidance in improving our manuscript.

The Figure 2 is controversial. On y axis there is "Probatility", but the values are below 0! Please have a think on it.

Thank you; we value your detailed review. You are correct that probabilities cannot be less than zero. The error occurred during the drawing and interpretation of the data. We corrected this mistake and included the updated figure in the manuscript.

It is also thought-provoking that results on original and modified data are different. According to my experience, in case of models based on tree (i.e. XGBoost) the normalised measures should not be different for scaled and non-scaled data.

Thank you; we appreciate your insightful comments. While it is true that tree-based algorithms are generally not sensitive to data normalization, in cases where there is a significant difference in the range of different features, the results for normalized data may differ from those for non-normalized data. This can affect the choice of splitting thresholds, which is a characteristic observed in some of our data.

Submission Date

19 June 2024

Date of this review

23 Jul 2024 08:35:23

Round 2

Reviewer 1 Report

Comments and Suggestions for Authors

The author of the article has revised the article accordingly in response to the comments, but the introduction suggests combining some of the paragraphs for continuity of reading.

Comments on the Quality of English Language

English still needs further improvement.

Author Response

Comments 1: The author of the article has revised the article accordingly in response to the comments, but the introduction suggests combining some of the paragraphs for continuity of reading.

Response 1: Dear Reviewer.

Thank you very much for your insightful feedback. In response to your suggestion, we have revised the "Introduction" chapter to improve the continuity of the text by combining some of the paragraphs. We believe that these changes have improved the flow and readability of the manuscript. Your input has been invaluable in refining our work and we greatly appreciate your contribution.

Comments 2: Comments on the Quality of English Language

English still needs further improvement.

Response 2: Thank you for your comment regarding the quality of the English language in our manuscript. We apologize for any imperfections, as we are not native English speakers. To ensure clarity, we had the manuscript professionally edited by the authorized MDPI language editing service. We trust that the revisions have addressed the concerns. If there are still any issues, we may consider following up with the service to ensure the highest quality.

We sincerely appreciate your valuable feedback and your understanding.

Reviewer 3 Report

Comments and Suggestions for Authors

Dear Authors,

Below you will find my comments after reading new version (v2) of your manuscript.

1. Paper structurization

The structurization got better in the version v2.

2. References

I understand your standpoint for references. I recommend grouping the references in section 4.4 into two groups: (1) using ML methods in quality inspection and (2) using ML methods for other purposes with regard to binary classification. It will be more clear for reader and you will achieve your goal to present ML methods in broader context.

3. Input data

OK, I see.

4. Materials and methods

You wrote as the answer to my remark: (...) we revised the text in Section 4 to provide a more detailed explanation of the reasons for choosing the models used in our study. Additionally, we clarified that the models were indicated by AutoML

Sorry, but I can't see such elements in manuscript. If I am wrong - can you indicate me the lines with such "detailed explanation"? In previous version the choosing the models using AutoML was mentioned in the abstract only. Now the abstract was changed and this information disappeared from the article. Please update it in the body of the article.

5. Results

What concern your remark: (...) it is true that tree-based algorithms are generally not sensitive to data normalization, in cases where there is a significant difference in the range of different features, the results for normalized data may differ from those for non-normalized data. This can affect the choice of splitting thresholds, which is a characteristic observed in some of our data.

Please add this explanation to the manuscript with the appropriate reference(s) that indicate(s) this property of tree-based models (that's why I marked "Are the results clearly presented?" as "Must be improved")

I would like to address also some new issues that appeared in version v2 of the manuscript.

In the "Introduction" there are a lot of "informal subsubchapters" ("The Background of the Study", "The Value and Significance of this Study", etc.). I think it is not necessary to include them in the manuscript, because it clearly stems from the content of the "Introduction".

You also stated three Research Questions in the "Introduction". In my opinion it is not necessary in your article (it's the results of response to other reviewers' comments, I suppose), but if you states such elements in your article, these Research Questions should be clearly addressed in the conclusions of your article (how you answered these question in your research?). Your article lacks of it. Please update it (that's why I marked "Are the conclusions supported by the results?" as "Must be improved") or remove these Research Questions from your article (in my opinion there are not necessary in case of your article).

Author Response

Comments 1:

  1. Paper structurization

The structurization got better in the version v2.

  1. References

I understand your standpoint for references. I recommend grouping the references in section 4.4 into two groups: (1) using ML methods in quality inspection and (2) using ML methods for other purposes with regard to binary classification. It will be more clear for reader and you will achieve your goal to present ML methods in broader context.

Response 1:

Dear Reviewer.

Thank you very much for your insightful suggestion regarding the grouping of references in section 4.4. We truly appreciate your feedback and have made the necessary adjustments to the manuscript. The references are now organized into two distinct groups at the end of section 4.4: (Group 1) those using ML methods in quality inspection, and (Group 2) those using ML methods for other purposes related to binary classification. We believe this restructuring enhances the clarity of our discussion and provides the broader context we intended.

We are grateful for your valuable input, which has helped us improve the quality and readability of our manuscript.

Comments 2:

  1. Input data

OK, I see.

  1. Materials and methods

You wrote as the answer to my remark: (...) we revised the text in Section 4 to provide a more detailed explanation of the reasons for choosing the models used in our study. Additionally, we clarified that the models were indicated by AutoML

Sorry, but I can't see such elements in manuscript. If I am wrong - can you indicate me the lines with such "detailed explanation"? In previous version the choosing the models using AutoML was mentioned in the abstract only. Now the abstract was changed and this information disappeared from the article. Please update it in the body of the article.

Response 2:

Thank you sincerely for your valuable feedback and for bringing this to our attention. We deeply apologize for the oversight—indeed, an important component regarding AutoML was inadvertently omitted from the manuscript. Your remark has been incredibly helpful in identifying this gap.

We have now updated the manuscript to include the relevant information about AutoML in the "Introduction" chapter.

We are truly grateful for your diligence and for helping us improve the quality of our manuscript.

Comments 3:

  1. Results

What concern your remark: (...) it is true that tree-based algorithms are generally not sensitive to data normalization, in cases where there is a significant difference in the range of different features, the results for normalized data may differ from those for non-normalized data. This can affect the choice of splitting thresholds, which is a characteristic observed in some of our data.

Please add this explanation to the manuscript with the appropriate reference(s) that indicate(s) this property of tree-based models (that's why I marked "Are the results clearly presented?" as "Must be improved")

Response 3:

We greatly appreciate your thoughtful feedback and the time you have taken to provide such valuable suggestions.

In response to your comment, we have added the suggested explanation to the manuscript in the Results section.

We believe this addition clarifies the issue and enhances the overall understanding of the manuscript.

Thank you once again for your insightful feedback, which has significantly contributed to improving the quality of our work.

Comments 4:

I would like to address also some new issues that appeared in version v2 of the manuscript.

In the "Introduction" there are a lot of "informal subsubchapters" ("The Background of the Study", "The Value and Significance of this Study", etc.). I think it is not necessary to include them in the manuscript, because it clearly stems from the content of the "Introduction".

You also stated three Research Questions in the "Introduction". In my opinion it is not necessary in your article (it's the results of response to other reviewers' comments, I suppose), but if you states such elements in your article, these Research Questions should be clearly addressed in the conclusions of your article (how you answered these question in your research?). Your article lacks of it. Please update it (that's why I marked "Are the conclusions supported by the results?" as "Must be improved") or remove these Research Questions from your article (in my opinion there are not necessary in case of your article).

Response 4:

We appreciate your detailed feedback and valuable suggestions. Based on your comments, we have removed the "Research Questions" section from the "Introduction" to better streamline the content and focus on the core issues of the paper. However, we have retained certain elements of the Introduction to ensure that we adhere to the formal structure required by the MDPI template.

We believe that these revisions have improved the clarity and coherence of the manuscript. Your guidance was instrumental in this improvement, and we are grateful for your continued support.

Round 3

Reviewer 3 Report

Comments and Suggestions for Authors

The manuscript can be accepted in the present form.